


# Dynamicity of multi-channel rip currents induced by rhythmic sandbars

Yao Zhang[a], Xiao Hong[b], GuoDong Xu[a], Xunan Liu[a],

Xinping Chen[a], Yuxi Sun[a], Bin Wang[a,*], Chi Zhang[c]

(a) National Marine Hazard Mitigation Service, Ministry of Natural Resources, Beijing, China, 100000

(b) South China Sea Prediction Center, Guangzhou, China, 510000

(c) Hohai University, Nanjing, China, 210098.

(*) Corresponding Author, email: mnr_wangbin@outlook.com

**Abstract**

In response to frequent fatal beach drownings, China's first operational attempt on the rip current hazard

investigation was made by the National Marine Hazard Mitigation Service (NMHMS). A great number of

recreational beaches were found developing rip currents interlaced with rhythmic sandbars, varying by season

and location evidenced by satellite images and morphodynamic calculation. Considering insufficient under-

standing of the multi-channel rip system, case analysis and numerical study were conducted to explore its

dynamicity and circulation characteristics under various wave climates in present work. The strength of rip

currents was generally proportional to wave height and channel width under certain limits. Increasing wave

height was not always a promotion and could even weaken the rip current due to the strong wave-current

shear. Interesting "pump" and "feed" interactions between adjacent rip currents in the multi-channel system

were observed. The rip current might be totally absent in narrow channels when the majority of water flows

through neighboring broader pathways. The rip current was highly sensitive to the incident wave angle.

Alongshore currents prevailed over the rip current when the wave angle reached 11 degrees to shore normal,

which was not favorable to the existence of channeled sandbars. Vortices appeared around the edge of the bar

owing to nonuniform wave breaking over rapid-varying bathymetry. The setup water was created shoreward

by the sandbar array and substantially increased as the wave deviated from the normal incidence. The water

surface depression in the rip channel was not observed as the wave angle increased, which fundamentally

explained why the rip current could not persist when the incident wave became slightly oblique. In future,

incident wave angle should be further incorporated into empirical formulas or probabilistic models to predict

the rip current for expected improvement in accuracy.

Keywords: rip current, rhythmic sandbar, wave angle, multi-channel, modelling



# Technical Background

The rip current is a narrow, fast-moving, seaward flow driven by the radiation stress and pressure gradient from the combined effect of nearshore wave and topography. It is likely to develop in submerged channels at sandy coast, where the setup water brought by consecutive breaking waves returns through the concentrated path. A rip flow may reach up to hundreds of meters long and meters-per-second fast, initiated at the vicinity of the shoreline and all the way through the surf zone. The location, flow pattern, and strength vary by topographic and hydrodynamic conditions, and could impact conversely. Featuring relative calm surface, the rip current could be deceptively powerful and quickly sweep incautious swimmers of all ability levels into deep water (Drozdzewski et al., 2012). According to NOAA's statistics in Figure 1, the rip current has been the primary littoral hazard in the United States, causing approximately 60% of surf zone fatalities in recent five years (Brewster et al., 2019). It also creates troubles to beach safety in Europe, Australia, Asia, and many other regions (Macmahan et al., 2006; Brighton et al., 2013; Kumar and Prasad, 2014; Arozarena et al., 2015; Barlas and Beji, 2016; Linares et al., 2019). Therefore, research efforts to protect people from the rip current hazard have experienced significant progress and mainly include mechanism study, direct observation, and risk prediction (Dalrymple et al., 2011; Castelle et al., 2016a).

Experimental and numerical modelling are sophisticated approaches to better understand the flow structure and characteristics of rip currents. Scaled experiments carried out in wave basins could explicitly demonstrate the hydrodynamic process over specific shoal-channel topography for various wave-tide environment, which is mathematically difficult to address. Fixed or movable bed may be applied to examine the interaction between hydrodynamics and morphodynamics (Castelle et al., 2010). The properties are measured by drifters, wave gauges and velocimeters. However, some shortcomings such as narrow-banded spectra of wave generation, inevitable scale effect, and high laboratory cost hinder studies for a large number of varying setups (Haller et al., 2002; Kennedy and Thomas, 2004; Kennedy and Zhang, 2008; Hur et al., 2019). As a complementary approach, the numerical model is able to compute long time series of water surface deviations and velocity profiles through many sets of simulations for varying bathymetric and hydrodynamic conditions (Long and Tuba, 2005; Suanda and Feddersen, 2015; McCarroll et al., 2015). Instead of narrow-banded spectra, the numerical model can simulate a broad range of wave spectra once mathematically specified (Weir et al., 2011; Marchesiello et al., 2015; Zhang et al., 2014). Boussinesq-type models provide the phase resolving approximation of highly nonlinear and dispersive wave-current dynamics with good computational efficiency


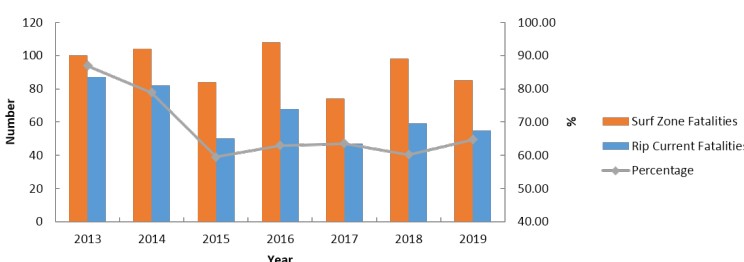

Figure 1: NOAA's surf zone fatalities statistics (https://www.weather.gov/safety/ripcurrent-fatalities)

<sup></sup>57   and accuracy (Zhang et al, 2013). The nonhydrostatic correction to the pressure and the good representation

of depth-varying velocities make them ideal tools for the numerical study of the rip current (Chen et al.,

1999; Haas et al., 2003; Johnson and Pattiaratchi, 2006; Zhang et al., 2016; Liu et al., 2018).

Alternate straightforward study of rip currents is the observation by either optical images or field equip-

ment. Satellite images are convenient source to directly identify rip currents as dissected narrow dark water

between sandbars or discontinuous white-cap breaking waves for a broad range of coastlines, but with limited

acquisition times (Lippmann and Holman, 1989; Athanasiou et al., 2018). Aerial or fixed optical cameras

are able to capture rip currents through consecutive observations covering long enough time and large scope

of view (Stockdon and Holman, 2000; Yoon et al., 2014; Liu and Wu, 2019). The size of the rip could be

extracted by either the visual or computational interpretation (Benassai et al., 2017). In the field observation,

the low-cost colored dye tracer or GPS drifter has been widely adopted to depict the two-dimensional rip

pattern by recording the speed and trajectories for relative wide spatial and temporal range in the surf zone

(Schmidt, 2003; Austin et al., 2010; Brander et al., 2014; Lumpkin et al., 2017). In addition, fixed-position

deployment of flowmeters allows the collection of flow velocities through specified cross section or depth, which

is instrumental in estimating the variability of the rip current for long time-series (MacMahan et al., 2005;

Mccarroll et al., 2014; Scott et al., 2016). Boat mounted ADCP, Airborne LiDAR, and X-band radar are all

effective in providing direct or resolved information of the rip current (Radermacher et al., 2018; Holman and

Haller, 2013; Haller et al., 2014).

Due to the intrinsic instability of the rip current (Yu and Chen, 2015), predicting exactly when and

where a rip current would occur is currently not feasible. Therefore, empirical analysis and assessment based

on long-time statistics has become an alternative avenue, forming two major methods: morphodynamic

beach classification and probabilistic forecast models. The morphodynamic method calculates dimensionless




sediment fall velocity $\Omega$ and tide-wave parameter $RTR$, which divide highly diverse beach states into 8-9 types

featuring different levels of rip risk in response to environmental conditions like wave, tide, and sediment.

The $\Omega$ indicates the mobility of the sediment which subsequently affects the nearshore morphology. Its large

value means the dissipative beach with high-level wave energy and fine sediments, while the small value

represents the reflective beach with coarse sediment and small wave. On the other hand, the $RTR$ compares

the contributions between wave and tide, which distinguishes the concave shoal under high-energy waves from

the flat tidal terrace. The method was proposed very early by Wright et al. (1985) and then improved during

its application to risk assessment for rip currents at various beaches (Masselink and Short, 1993; Scott et

al., 2011; Li, 2016). The probabilistic model is based on the logistic regression of statistical data of wave

conditions and water level to predict the occurrence probability of rip currents (Alvarezellacuria et al., 2010;

Moulton et al., 2017a). Coupled with wave and circulation models, the probabilistic model has been made

as an operational forecasting system for the rip current risk by the NOAA National Weather Service ( Dusek

and Seim 2013; Dusek et al., 2014; Churma et al., 2017). Although these approaches are only capable of

providing limited quantitative information, the computed likelihood of hazardous rip currents is still very

useful.

## Motivation

China's coastal tourism contributes 46.1% to domestic marine economy (China Marine Economic Statistics

Bulletin, 2018). Unsurprisingly, some favored beaches have been frequently reported dangerous rip currents

and deadly drownings, which unfortunately has not been widely recognized. Surfers and swimmers sometimes

take rip currents as the "express lane" to expedite entering deep water, which is highly risky even with robust

experience and extreme caution. The public, media, local administration, even specialists have vague concepts

about the rip current. Not too much research effort has been made on the rip current by China's scientist

community yet. As a huge negative effect, the rip current hazard prevention in China lags behind but has been

imperative. The National Marine Hazard Mitigation Service (NMHMS), Ministry of Natural Resources, has

launched a nation-wide rip current investigation at China's major coastal recreational beaches, in responding

to frequent deadly drowning accidents.

It was found that a great number of beaches in China develop alongshore rhythmic sandbars dissected

by rip channels, similar to those in United States, Europe, Australia, and other regions (Houser et al., 2011;





Ribas et al., 2015). The littoral morphology and rip current vary by locations and show obvious seasonal

variations evidenced by satellite images and the morphodynamic calculation. Although the comprehensive

risk assessment of rip currents could be properly related to long-term observation statistics (Scott et al.,

2014), the evolution and mechanism of this multi-channel rip system still remains insufficiently understood

(Castelle et al., 2016b). Extensive previous research was focused on artificial topography with only one or two

idealized rip channels (Haller et al., 2002; Haas et al., 2003; Moulton et al., 2017b; Hur et al., 2019). Present

paper explores the dynamicity of multi-channel rip currents interlaced with rhythmic sandbars. Based on

the investigation, case study and numerical simulations were performed to check the sensitivity of the rip

circulation to incident wave conditions and topography. The study results fundamentally explain how the rip

current responds to the morphodynamic and hydrodynamic environment changes, which could be useful to

determine when and where sandbar-induced rip currents would develop.

## Case Study

In the rip current investigation, the morphodynamic beach state classification (Masselink and Short, 1993;)

and the satellite image interpretation were adopted to assess the occurrence of sandbar development and rip

currents at target recreational beaches. After that, labor-intensive field surveys were conducted to quantita-

tively observe rip currents at those identified shorelines. The results show that the rip current vary widely

by region and feature seasonal shifts. Preliminary estimated, southern coast shows higher statistics in both

frequency and intensity of rip currents compared to the north, due to the large wave and dynamic littoral

processes.

Figure 2 compares six satellite images taken at cool and warm seasons for three most visited beaches

in southern China mainland. The three beaches are distant from each other with different curvatures and

exposure orientations as shown in Figure 3. The coastal topography and sea condition in the left three photos

were relatively mild while there developed obvious sandbars or rip channels in the right three photos, which

indicates irregular temporal variations at the same beach and distinct characteristics of rip development for

different beaches. The red line in the image measures 100m of the beach cross section making sure observations

at similar tidal time were compared. Photo (b) of the Xichong Beach taken in summer shows the whitecap

of the breaking wave was noticeably pushed seaward by rip currents at gaps between alongshore sandbars,

with sediment transferred along with the outflow. In contrast, the break line of the wave was continuous

Natural Hazards
and Earth System
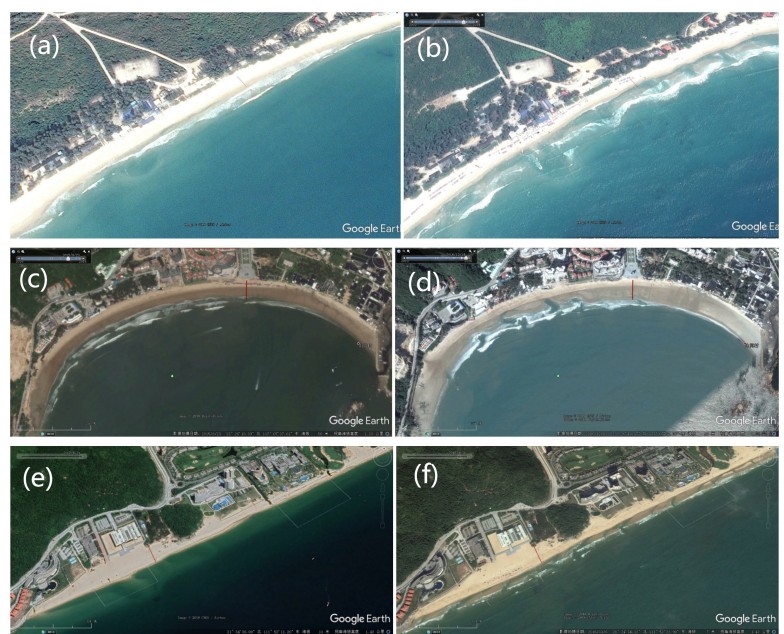

Figure 2: Satellite Images: (a) Xichong Beach, December 30, 2013; (b) Xichong Beach, August 25, 2015; (c) Qingao Bay, August 23, 2015; (d) Qingao Bay, December 18, 2016; (e) 10-mile Beach, August 25, 2015; (f) 10-mile Beach, November 6, 2016. © Google Earth.

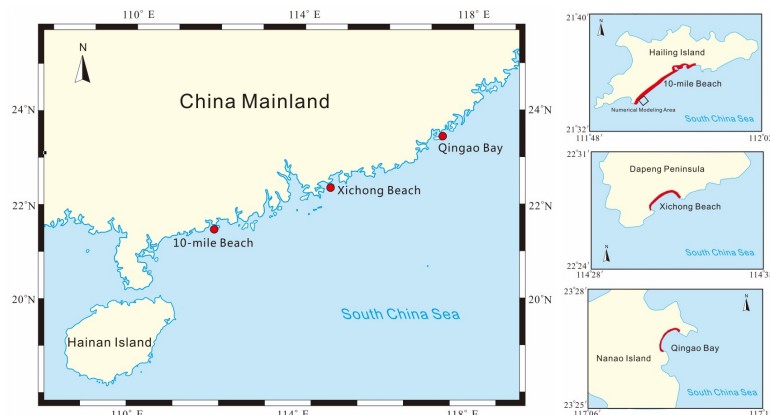

Figure 3: Locations and profiles for the three investigated beaches along south coast of China mainland.


in the winter photo (a) without any obvious sandbars or rip channels at the Xichong Beach. A probable

contraseasonal change of the rip current development was observed in satellite image at the Qingao Bay,

opposite to a popular misconception that hotter summer beach generates more rip currents. Different from

long straight beach, the beach of Qingao bay is a typical curved pocket beach which usually features high

risk of mega rips. There appeared long rip channels concave in the beach at the immediate vicinity of the

shoreline in the winter image (d) while a uniform flat beach and shoal without sandbars or channels were

displayed in the summer image (c). The satellite photos (e) and (f) of the 10-mile Beach as well show similar

dynamic evolution of rhythmic sandbars with those observed at the Xichong Beach. Rip channels truncated

quasi-linear alongshore sandbars as shown in summer photo (f) while the gentle underwater shoal was seen

through the calm water in winter photo (e).

Resembling the "butterfly effect", the rip current is extremely sensitive to environmental factors such

as tide, wave, topography, sediment, breaking, turbulence etc. Sandbars and rip channels are reshaped

ceaselessly by coastal hydrodynamics (Zheng et al., 2014; Lee et al., 2019), consequently resulting in very

dynamic rip circulations. In the case of little difference in tidal range and sediment condition, varying local

wave climate towards beaches with different profiles could be the main reason for this temporal and spatial

shifts of the rip current. Figure 4 shows the monthly averaged significant wave heights and wave periods in

recent two years collected from nearest stations or buoys. Xichong beach and 10-mile beach shared similar

trends in temporal changes of both wave height and period, with incident directions as SSE in summer and

ESE in winter. This might explains their similarity in the development of rhythmic sandbars, although the

wave period at 10-mile beach was slightly smaller than that at Xichong Beach. However, the wave climate at

Qingao Bay was a totally different. Compared to the other two beaches, the seasonal variation in the wave

height showed contrary undulation with larger amplitude, while the wave period was much smaller. The

0.8m-high wave in winter and 0.4m-low wave in summer hint a probable seasonal shift of the rip current. It

is worth noting that the seasonal difference of the wave height may be correlated to the eastward bay mouth

where incident wave directions are SSW-SE in summer and ENE-E in winter.

Field surveys were carried out for the verification of the local rip current condition. The strength and flow

pattern of rip currents were estimated by the airborne camera and colored dye tracer. Figure 5 demonstrates

two aerial photos of rip currents at the 10-mile beach on July 11, 2018, capturing rip currents induced

by discontinuous crescentic sandbars. "Calm" rip currents were observed between white-cap breakers over

sandbars in the left photo, while the visual characteristics of the rip current in the right photo included

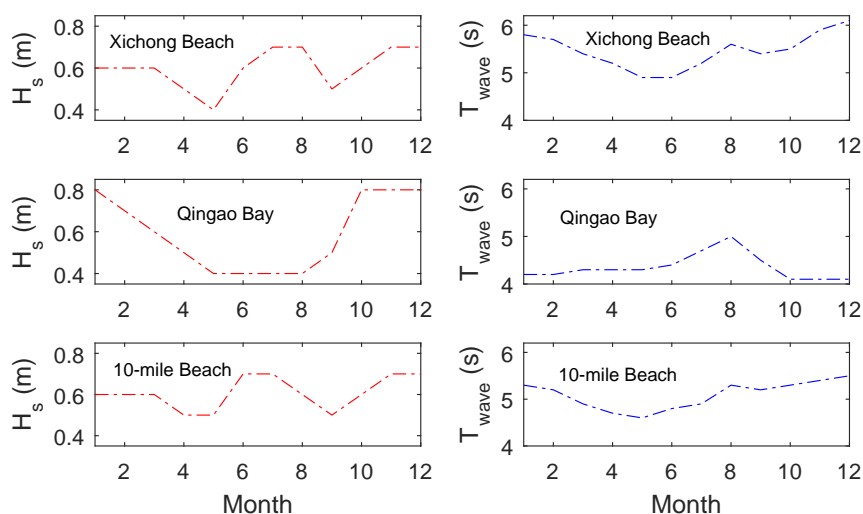

Figure 4: Monthly averaged: significant wave heights and wave periods at the study region

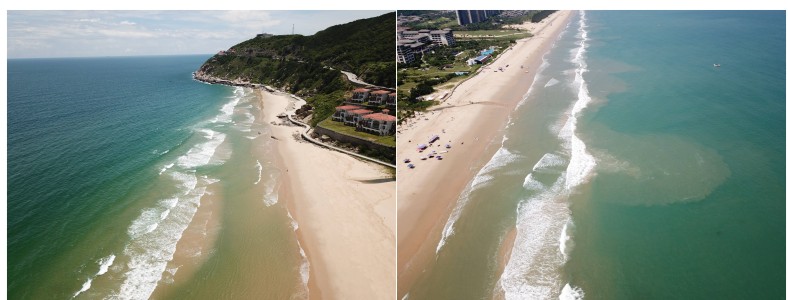

Figure 5: Aerial photo of bar-induced rip currents and a rip head at the 10-mile beach.

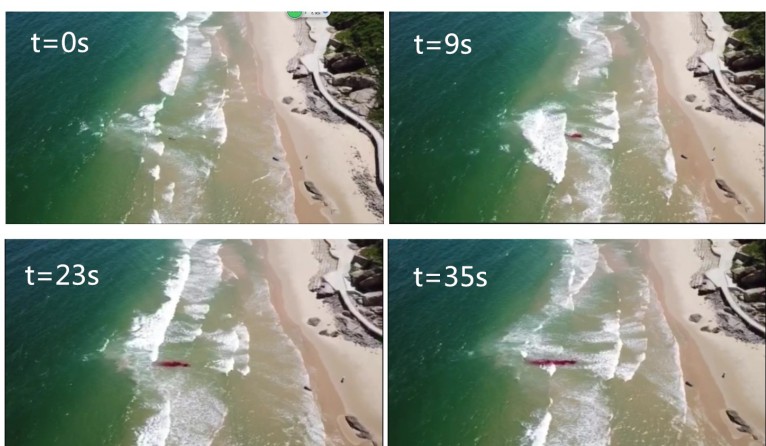

Figure 6: Video snapshots of the dye-tracing experiment capturing a rip current at the 10-mile beach.

foam and sediment drifting offshore into a giant "rip head". It was verified in the field survey that the rip

current was more easily developed or got stronger toward the low tide when shallower water intensified the

topographic effect on waves and currents, which was in accordance with literatures and lifeguards' experience

(Bruneau et al. 2009; Austin et al., 2014). It is reasonable that higher wave energy would be required to

drive the thicker water layer than the lighter water mass. Figure 6 shows video snapshots of a dye-tracing

field experiment conducted at the 10-mile beach on July 11, 2018, in order to quantify the velocity magnitude

of the rip current. Local significant wave height was 1.4m, doubled July's monthly averaged value 0.7m, with

a SE-SSE direction which is nearly perpendicular to the shoreline. Acid Red 14 was selected as it remains

clearly visible while diluted and spread. The dye release spot deviated quite a bit from the center line of rip

current, capturing a moderate stable flow speed around $0.6m/s$ to roughly calibrate the following numerical

modelling.

## Numerical Study

Given the variability appeared in the investigation, numerical study was carried out to further check circu-

lation characteristics of the multi-channel rip currents on rhythmic-sandbar topography under various wave

conditions. Figure 7 shows the bathymetry of the $1km - 1km$ simulation domain at the same place where

the previously described dye experiment was conducted (channel 2). The white margin stands for the land

area with elevation above 0m. Quasi-linear underwater sandbars are clearly seen offshore with their length



Natural Hazards
and Earth System
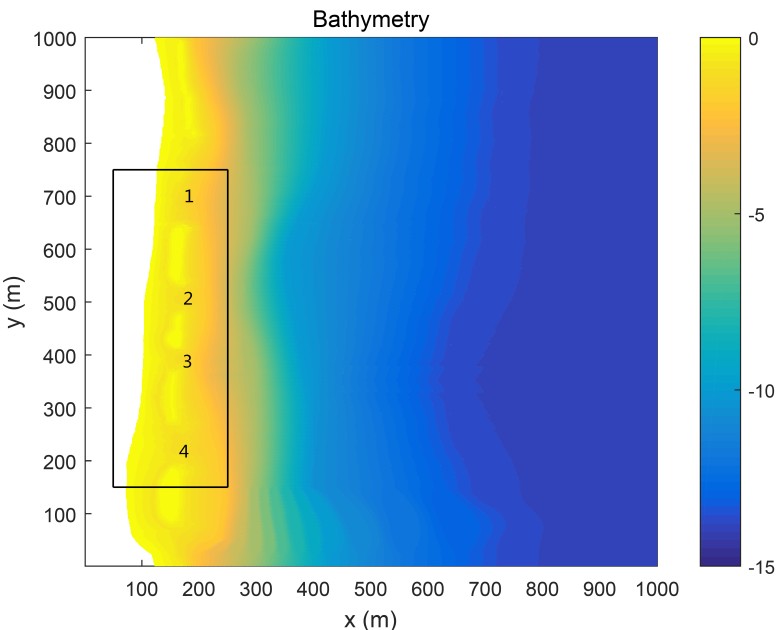

Figure 7: Bathymetry of the simulation domain with rhythmic sandbars and rip channels

range from 70 to 170 meters. The widths of rip channels labeled 1-4 are respectively 160m, 52m, 38m, and

82m. Boussinesq phase-resolving model, FUNWAVE (Shi et al., 2012), was used for a group of rip current

simulations for different wave heights and incident directions, applying the 1m uniform grid size. Results of

six selected test cases, as shown in table 1, are presented in the following section. The simulation time for

each case was 50 minutes with a 50-second output interval, long enough to avoid the startup effects and reach

stable states, while for some statistics to be obtained.

Figure 8 compares the averaged velocity field and the circulation pattern for increasing significant wave

heights from 0.7m to 2.0m at normal incidence (Test 1-4). The flow regimes and characteristics at the four

rip channels were completely different. Straight-seaward rip currents at channel 2 and 4 were distinctly

developed, while a semi-circular flow consisting of an oblique arc-shape rip and the feeder current behind the

top sandbar was generated at the lower side of channel 1. Rip currents were barely observed at channel 3

for either small or large wave conditions. The strength of rip currents appeared proportional to the channel

width for channel 2, 3, 4, except channel 1 which is too wide to concentrate the flow. The velocity and the

decay length increased with the increasing incident wave height from 0.7m to 1.4m, maintaining the flow

pattern unchanged. However, for $H_i = 2m$, the large incident waves turned the rip into a complete eddy




Table 1: Incident wave conditions for the numerical simulations

| Test No. | Sigificant Wave Height (m) | Direction | Period (s) | Propagation angle to lateral boundary (°) |
|---|---|---|---|---|
| 1 | 2.0 | SE-SSE | 4.9 | 0 |
| 2 | 1.4 | SE-SSE | 4.9 | 0 |
| 3 | 1.2 | SE-SSE | 4.9 | 0 |
| 4 | 0.7 | SE-SSE | 4.9 | 0 |
| 5 | 1.4 | SSE | 4.9 | 11.25 |
| 6 | 1.4 | S | 4.9 | 33.75 |

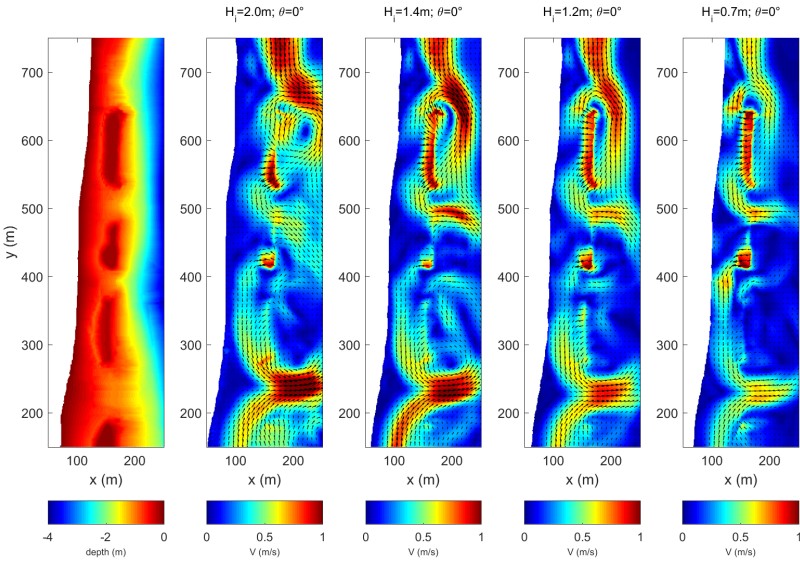

Figure 8: Computed spatial distribution of averaged velocities for different incident wave heights (0°).


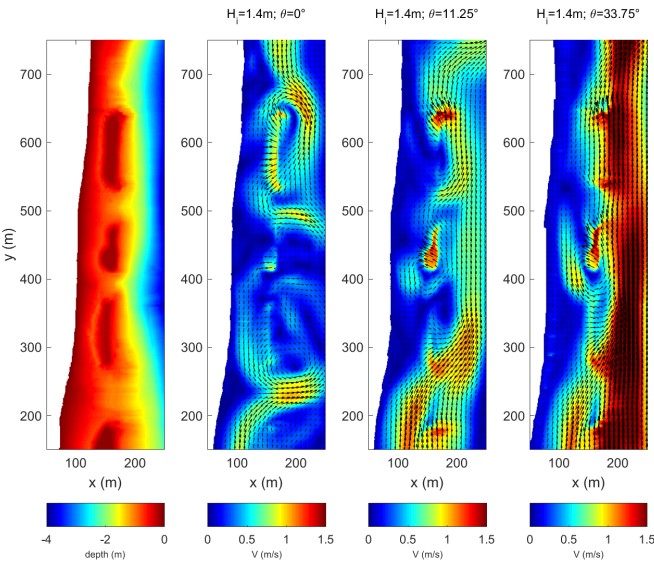

Figure 9: Computed spatial distribution of averaged velocities for different incident angles ($Hs = 1.4m$).

loop outside of channel 1 and dispersed the rip current at channel 2, due to strong wave-current shear in

the offshore velocity profile; Meanwhile the rip current at channel 4 still got strengthened in proportion to

the wave height. At the 10-mile beach, the rip current could reach a maximum averaged speed of 0.7m/s

for July's averaged wave condition or around 1m/s on that experimental day. Interesting 'feed' or 'pump'

interactions between adjacent rip currents were as well observed. Not only for self-circulation, the outflow

from channel 1 served as a feeder current across the sandbar for the rip current at channel 2. On the other

side, an alongshore current behind sandbars originated from the upper side of channel 3 flowed all the way

into the rip current at channel 4, forming one of the two stable feeders. This was why no obvious outflow

went through channel 3. Therefore, in a multi-channel case, the rip current might be totally absent in small

channels when majority of water flows out through neighboring broader pathways.

In addition to wave height, few studies have paid attention to the rip's sensitivity to incident wave angle

(Moulton et al., 2017). Here, averaged velocities, vorticity, and water setup for $H_s = 1.4m$ with varying wave

propagation directions were computed and analyzed (Test 2, 5, 6). Figure 9 shows rip currents deflected

into undulant alongshore currents in front of the sandbars as the wave angle increased. The 11.25 degree

was identified as the threshold of the incident direction when the alongshore current prevailed over the rip

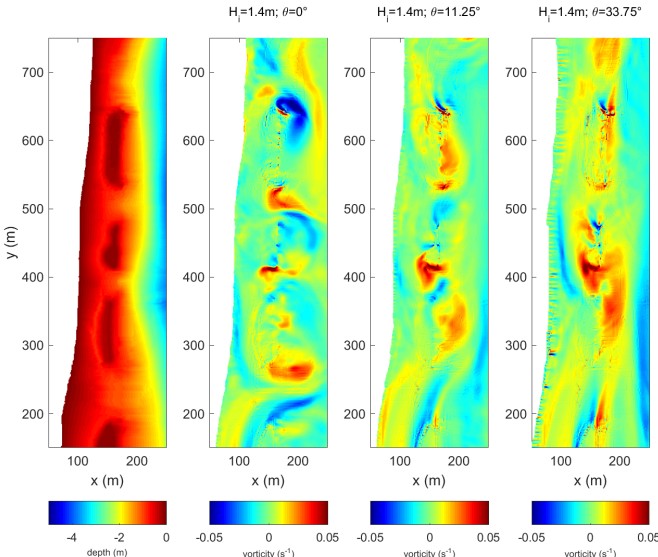

Figure 10: Computed spatial distribution of averaged vorticity for different incident angles ($Hs = 1.4m$).

current. When the wave angle exceeded 30 degrees, the oblique incident wave generated strong alongshore

currents with the speed reaching 1.5m/s, while the velocity in the bar-shielded area was relatively much

smaller. Although rip currents no longer persisted when the incident wave became oblique, a circular flow

was formed at the junction of channel 3 and its upper sandbar. Hence, waves even with the same wave height

and period could drive nearshore currents with great differences in the flow pattern when propagation angles

were slightly different. The results are expected to be further discussed and checked by researchers in future

work.

The computed vorticity provided an alternate perspective to see the influence on the rip current by the

changing wave direction. The vorticity is defined as $\omega = v_x - u_y$ in $s^{-1}$, where $(u, v)$ were averaged current.

Figure 10 plots the spatial distribution of the averaged vorticity for varying wave propagation angles with

the identical $1.4m$ significant wave height. For the normal incidence, stretched vortex pairs were clearly seen

along the rip current at channel 1, 2, 4, while some concentrated vorticity balance appeared at the edge of

sandbar. As the wave angle increased, those vortex pairs disappeared with spins of the flow arising near

sandbars, which corresponds to the observation in previous velocity plots.

Figure 11 shows the computed mean water elevation at the sandbar area for the same wave height with

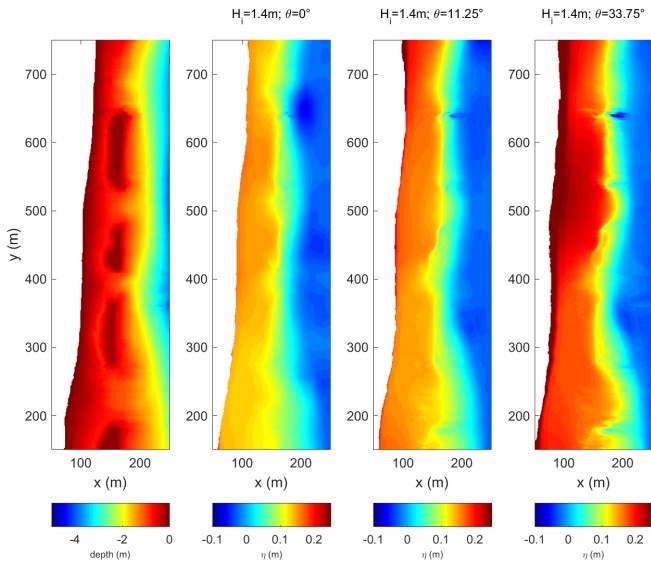

Figure 11: Computed spatial distribution of averaged surface elevation for different incident angles ($Hs = 1.4m$).

different incident directions. The sandbar array created the water setup shoreward and the setdown in front

of its outer edge, which was very similar to the hydraulic jump. The setup elevation increased with the wave

angle significantly. For $\theta = 0°$, the gradient of the water level was clearly seen in rip channels because of the

nonuniform wave breaking on the rapid-varying bathymetry, which provided pressure differential to drive the

feeder flow toward the channel. However, the surface depression in channel 1, 2, 4 was not observed as the

wave angle increased, while it is usually expected in a channeled rip system (Zhang et al., 2016; Bertin et

al., 2018). This may explain from the mechanism why rip currents vanished when the incident wave angle

became slightly oblique.

## Discussion and Conclusion

Present work, largely the extended research of China's nation-wide rip current investigation, studied the dy-

namicity of multi-channel rip currents generated by rhythmic sandbars. The investigation adopted multiple

methods such as the widely-used empirical beach state classification formulas, remote sensing image interpre-

tation, and labor-intensive field survey. Many recreational beaches in China were found developing rhythmic




sandbars and hazardous rip currents with temporal & spatial variations evidenced by satellite images and the

morphodynamic calculation. For three demonstrated beaches, this variability showed strong correlation to

the changing wave climate and the beach profile. Beside the observed wave height, the contrasting exposure

orientation of Qingao Bay to the incoming wave suggests potential influence of the wave direction on the rip

current. In accordance with literatures, it was confirmed in the field observation that the rip current was

most hazardous at low tide when shallower water intensified the topographic effect on waves and currents.

Dye release under the aerial recording is a practical approach to quantify the speed of the rip current.

Numerical simulations quantitatively revealed the dynamicity of the multi-channel rip currents on rhyth-

mic sandbar topography under various wave conditions. The nearshore circulation pattern was sensitive to

the geometry of sandbars and channels. The strength of rip currents was generally proportional to wave height

and channel width under certain limits. However, increasing wave height would not always be a promotion

to the rip current and may even smash the rip due to the strong wave-current shear. Interesting "pump" and

"feed" interactions between adjacent rip currents in a multi-channels system were observed. Therefore, the

rip current might be totally absent in small channels when the majority of water flows through neighboring

broader pathways.

The incident wave angle was found a significant impact factor on the rip current in the computation.

Rip currents gradually deflected into alongshore currents in front of sandbars as the wave deviated from the

normal incidence. Alongshore currents dominated over the rip current when the incident wave angle reached

$11°$, which was favorable to alongshore-uniform evolution and the disappearance of sandbar channels. Thus,

not just the wave direction, the incident wave angle to the target shoreline should be incorporated into

existing empirical formulas or probabilistic models to improve the accuracy of the rip current prediction.

Circular flow usually appeared at the bar-channel junction because of nonuniform wave breaking over rapid-

varying bathymetry (Bruneau et al, 2011). The setup water was created shoreward by the sandbar array

and substantially increased with the wave angle. The water surface depression in the rip channel was not

observed as the angle increased, which fundamentally explained why the rip current could not persist when

the incident wave became slightly oblique.

The scientific findings provide fundamental advance in the understanding of coastal currents and could

be useful to enhance the beach recreational safety. In the future, long-time observations for specific sections

of the shoreline would be carried out to accumulate enough statistics for rip current prediction and risk

governance.



## Acknowledgments

This work was funded under the operational funding of National Marine Hazard Mitigation Service, Ministry of Natural Resources, P.R. China and the National Natural Science Foundation of China grant 51609043, 51879096. Their support is gratefully acknowledged. We would like to extend our appreciation to relevant literatures that greatly helped China's rip current investigation which may have already saved many lives.

## Author Contribution Statement

Y.Z. led the investigation and the research, wrote the main manuscript. X.H. participated in the investigation, visualized the numerical simulations. G.X. participated in the investigation, reviewed the manuscript. X.L. contributed to the satellite image interpretation. X.C. participated in the investigation. Y.S. and C.Z. contributed to the data analysis and the manuscript writing. B.W. proposed the methodology of the investigation and the research, reviewed the manuscript.

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

159.