# Peer review of "Dynamicity of multi-channel rip currents induced by rhythmic sandbars"

_Natural Hazards and Earth System Sciences, 2020_

## Short Comment (SC1) · 3 Jun 2020

It's a very decent manuscript on the rip current. The extensive study reveal that the rip current is very sensitive to the incident wave angle. The authors also found that increasing wave height was not always enhance the rip current. Those are very import conclusions and can help to understand the mechanism on the generation of rip current.

---

## Referee Comment (RC1) · Anonymous Referee #1 · 9 Jun 2020

I have enjoyed reading the manuscript. It provides the characteristics of rip currents in China's beaches and related research in China.

I would suggest providing more information about differences in beach characteristics and hydrodynamics, such as wave climate and tides. It seems that the tidal range is generally large in China versus US. If that's true, the tide may have significant effects.

Figure 11 looks abnormal for me. Does the color represent wave setups? Did you use the same color coding for the four plots. Wave setup from the more oblique incidence looks larger than the normal incidence, which is opposite in terms of the momentum balance. The radiation stress component, Sxx, should decrease with increasing incident angle, and thus the pressure gradient decrease. Please refer to Hsu et al. 2006 Hsu, T.-W., Hsu, J.R.-C., Weng, W.-K., Wang, S.-K., Ou, S.-H., 2006. Wave setup and

[Figure]

setdown generated by obliquely incident waves. Coast. Eng. 53, 865–877.

Specifics:

Line 16: wave-current shear, wave-current interaction?

Line 80: "The $\Omega$ indicates the mobility of the sediment which subsequently affects the nearshore morphology. Its large value means the dissipative beach with high-level wave energy and fine sediments, while the small value  represents the reflective beach with coarse sediment and small wave." My question is why the settling velocity is larger for fine sediment? Please provide more info about RTR. . . Line 146: tide, wave, topography, sediment, breaking, turbulence. Suggest: topography, tide, wave breaking, etc. . Line 155: "was a totally different." rewording . Line 179 1km − 1km  . Suggest: 1km x 1km. . Line 228: which was very similar to the hydraulic jump.  It's not an appropriate analogy. Hydraulic jump is a discontinuity.
* * *

---

## Short Comment (SC2) · 11 Jun 2020

Very good research. In my opinion, it is an important finding that increasing wave height could even weaken the rip current due to the strong wave-current shear. This provides us with a potential solution to suppress the occurrence of rip current. The inspection of the coastal rip current is an effective guarantee for the safety of tourists. I hope this work will continue.

---

## Short Comment (SC3) · 21 Jun 2020

We appreciate the reviewer's instructive comments that would improve the quality and consistency of present manuscript.

It is a good suggestion to provide more comparative information for the case study, especially the tide. For the wave setup plot, we indeed used the same color scale and double checked the result. Nearshore morphological condition here was quite different to the smooth-slope bathymetry in Hsu's paper (2006) since the submerged sandbars significantly blocked the setup water from flowing back when the incident wave became oblique. Although it is a valuable wave setup literature that we missed, we recommend additional check of probable paradoxical results from Figure 5 and 7 in that paper.

[Figure]

The dimensionless fall velocity is defined as $\Omega=H\_b/(W\_s*T)$ while the tide-wave parameter is given by $RTR=TR/H\_b$. Therefore, the setting velocity is smaller for finer sediment. We would provide more information about these two parameter and fix those specific problems.

———————————————————

---

## Short Comment (SC4) · 23 Jun 2020

This is a highly valuable research. Based on multiple methods, this study reveals that wave height, channel width and incident wave angle have effects on the rip current. Meanwhile, this investigation indicates that the strength of rip currents is not always proportional to wave height. These findings are really interesting and very important for further understanding the rip current and implementing protection measures.
* * *

---

## Referee Comment (RC2) · Anonymous Referee #2 · 28 Jul 2020

General Comments This paper describes some measurements and numerical modelling relating to rip currents on three (although primarily one) beach locations in China. It is always useful to have scientific studies of rip currents in different wave climate environments so this study is useful. However, the authors make several strong statements that are not warranted. The methodology is very basic and limited and the field surveys are not described. The modelling is more robust. The primary findings support those of existing studies and this should be the main outcome of this study.

I have some doubts about the reliance of two different images of rip currents at these beaches based on 'seasons' that are only supported by two years of wave climate data. It is hard to know if the beaches are consistently different in morphology between seasons or if the conditions in the images were simply based on antecedent morphology.

[Figure]

There is a lot of information provided on beach safety, but no real implications to how the results of this study relate to beach safety in China.

The authors do a good job of writing in English, but there are still numerous examples of incorrect language and grammar, which is completely understandable. It would be useful if a colleague who has English as a first language could give it a good proof read.

I would suggest that the authors provide more detail on the beaches in China and how the beaches in this area are similar/different to other areas and describe the wave climate in more detail. They should also describe the methodology involved in their field surveys and tone down their methodology involving 'remote sensing' and the dye release, which only captures a snapshot in time of rip current speed and trajectory. While the authors do a good job of referencing literature, some of the classic papers on rip currents are not included and they should at least describe the types of rip currents present at their beach locations. There is some evidence of an incomplete understanding of the forcing mechanisms driving rip current flow, although this could be language/translation related.

I would suggest that the authors use their results to support existing findings and high-light what is new about their findings.

Some specific comments are provided below (some which relate to the general comments above): - Abstract L9 – a great number of recreational beaches? How many – this is not stated in the paper and the focus is only on 3 locations. - more detail on methods used is needed

Technical Background L29 – should be 'A rip current... L29 –L34 – these statements should have references to some of the classic rip current literature, the most recent review of which is by Castelle et al. 2016 (in the reference list) L63 – please note that Lippmann and Holman 1989 used remote video Argus cameras and not satellite images L69 – should also include the MacMahan et al. 2010 Marine Geology paper

in these references; the Brander et al. 2014 study did not use drifters but this one did: McCarroll, R.J, Brander, R.W., Turner, I.L., Power, H.E., Mortlock, T.R. (2014). Lagrangian observations of circulation on an embayed beach with headland rip currents. Marine Geology, 355, 173-188. L71 – many earlier papers that should be cited for measurements of rip flow using fixed current meters L73 – Radermacher references is not in chronologic sequence L80 – needs references to Wright and Short (1984) and Masselink and Short (1993) L97 – should reference this study: Li, Z., & Zhu, S. (2018). WhyThere Are So Many Drowning Accidents Happened at Dadonghai Beach, Hainan, China: Morphodynamic Analysis. Journal of Coastal Research, 741-745. L104 – is it possible to provide numbers or estimates of drowning accidents on recreational beaches in China? L117 – should add that this has implications for beach safety in China

Figure 1 is not necessary

Motivation - there is no real clear aims/objectives of the study; more detail is needed to support the last statement starting on L115

Case Study L121 – are the targeted beaches just the 3 shown in Figure 2 or were there more? Are the southern and northern coasts referred to just in the region of study shown in Figure 3? I would suggest not having the first paragraph but incorporate parts of it where appropriate in the following text. One problem I have is that it is not clear if the beach morphology changes seasonally consistently every year or if they only look different in the images because of normally variable wave conditions. Some more information should be given on the patterns of wave climate by season. L121 – can you describe what the 'labor-intesive field surveys' involved and when/how they were done? L132 – refer to Figure 2b rather than the photo (same for any other instances – refer to Figure) L145 – ok the wave climate information comes here, but perhaps wave climate should be described first before the rip current/beach morphology is described in the previous paragraph L150 – is 2 years enough to be confident about these seasonal trends – the two years should also go in the caption for Figure 5

[Figure]

L163 – white-cap breakers are just breaking wave activity, often known as 'whitewater' and the intensity will vary based on water depth (tides) and incident wave height. I don't really understand the 'calm' rip current term. These are channelized rip currents as defined in Castelle et al. 2016 L168 – there are many older studies that have shown the relationship between higher rip current flow at low tide L169 – here is where the Brander et al. 2014 paper on dye could be referenced; the dye release is not really an experiment. If it was just done once, it shows very little useful information. How was the velocity of 0.6 m/s obtained?

Numerical Study L179 – how was the bathymetry obtained? Why was this only done for the 10 Mile Beach location? Justification? Figure 7 caption should also explain what boxed areas and numbers (channels) refer to L185 – capitalise Table 1

Discussion and Conclusion L237- the rip currents are not generated by rhythmic sandbars, but are forced by temporal and spatial variability of wave breaking activity over the sandbars. The statement 'The investigation adopted multiple methods such as the widely-used empirical beach state classification formulas, remote sensing image interpretation, and labor-intensive field survey' is over-stated. It's not clear how the beach classifications were used (as a method), the remote sensing interpretation involved describing some Google Earth images and the field surveys are not described. L245- it was shown to flow faster at low tide, but 'hazard' depends on many other factors as well L266 – the statement 'The scientific findings provide fundamental advance...' is also too strong. Most of these findings support previous findings. I'm not sure what the advances are?

---

## Author Comment (AC1) · 8 Aug 2020

We greatly appreciate the reviewer's thorough reading and insightful comments which have largely improved the quality of present manuscript. We have carefully looked over all the remarks and made revisions stemming from those concerns and specific problems. The most significant changes arise from the inclusion of precise/objective statements, morphodynamic analysis, more field survey information, more related literatures, and more descriptions and discussions. In regard to the seasonal morphology difference, many remote images were collected (Fig. 1), from which only 2 of each beach were shown in the manuscript. The wave data has to be recent years to be representative and 2-year period should be enough for the open coast. As the reviewer suggested, we will have our manuscript proof read by an English-speaking colleague

to minimize the possible misunderstanding of the scientific content.

The major purpose of present work is to investigate the multi-channel rip circulation system and its dynamicity under season-varying wave conditions (especially wave directions), which is frequently observed in the real world without enough attention addressed. The quantitative scientific findings are expected to be useful to the understanding and precaution of rip currents. While present work focuses on the mechanism study, the larger scope of China's beach safety statistics, nation-wide rip current investigation and risk distribution will be introduced in future work which is undergoing (Fig. 2, 3).

Again, we totally acknowledge the reviewer's comments and as well extend our appreciation to the editor for the effort.

[Figure]

**Fig. 1.** Google Earth images at Qingao Bay on: (a), August 5, 2010; (b), November 27, 2013; (c), August 23, 2015; (d) December 18, 2016

**Fig. 2.** China's beach safety statistics from 2010-2019

**Fig. 3.** Risk distribution of the rip current at China's major recreational beaches, based on morphodynamic calculation, remote images, and field investigation

---

## Author Comment (AC2) · 8 Aug 2020

We appreciate the reviewer's instructive comments that will greatly improve the readability of present manuscript.

It is a good suggestion to provide more comparative wave and tide information in the case study. For the wave setup plot, we indeed used the same color scale and double checked the result. Nearshore morphological condition here was quite different to the smooth-slope bathymetry in Hsu's paper (2006) since the submerged sandbars significantly blocked the setup water from flowing back when the incident wave became oblique. It is a quite interesting wave setup paper that we had not previously been aware of. We have been reading it carefully to see how we might make use of some of

their concepts.

The dimensionless fall velocity is defined as $\Omega = H\_b/(TW\_s)$ while the tide-wave parameter is given by $RTR = TR/H\_b$. Thus, the setting velocity is smaller for finer sediment. We will provide more information about the morphodynamic analysis and fix those specific problems.

Thank you for the review.